# Antiviral Potential of Specially Selected Bulgarian Propolis Extracts: In Vitro Activity against Structurally Different Viruses

**DOI:** 10.3390/life13071611

**Published:** 2023-07-23

**Authors:** Neli Milenova Vilhelmova-Ilieva, Ivanka Nikolova Nikolova, Nadya Yordanova Nikolova, Zdravka Dimitrova Petrova, Madlena Stephanova Trepechova, Dora Ilieva Holechek, Mina Mihaylova Todorova, Mariyana Georgieva Topuzova, Ivan Georgiev Ivanov, Yulian Dimitrov Tumbarski

**Affiliations:** 1Department of Virology, The Stephan Angeloff Institute of Microbiology, Bulgarian Academy of Sciences, 26 Georgi Bonchev Str., 1113 Sofia, Bulgaria; inikolova@microbio.bas.bg (I.N.N.); nadyanik@yahoo.com (N.Y.N.); zdr.z1971@abv.bg (Z.D.P.); madi_trepechova@yahoo.com (M.S.T.); doraholechek@yahoo.com (D.I.H.); 2Institute of Morphology, Pathology and Anthropology with Museum, Bulgarian Academy of Sciences, 25 Georgi Bonchev, 1113 Sofia, Bulgaria; 3Department of Organic Chemistry, Paisii Hilendarski University of Plovdiv, 24 Tsar Asen Str., 4000 Plovdiv, Bulgaria; mm_todorova@abv.bg; 4Department of Organic Chemistry and Inorganic Chemistry, University of Food Technologies, 26 Maritsa blvd., 4002 Plovdiv, Bulgaria; marianagt@mail.bg (M.G.T.); ivanov_ivan.1979@yahoo.com (I.G.I.); 5Department of Microbiology, University of Food Technologies, 26 Maritsa blvd., 4002 Plovdiv, Bulgaria

**Keywords:** propolis extracts, antiviral activity, virucidal activity, viral adsorption, human coronavirus, human respiratory syncytial virus, herpes simplex virus, human rhinovirus, human adenovirus

## Abstract

Propolis is a natural mixture of resins, wax, and pollen from plant buds and flowers, enriched with enzymes and bee saliva. It also contains various essential oils, vitamins, mineral salts, trace elements, hormones, and ferments. It has been found that propolis possesses antimicrobial, antiviral, and anti-inflammatory properties. We have studied the antiviral activity of six extracts of Bulgarian propolis collected from six districts of Bulgaria. The study was conducted against structurally different viruses: human coronavirus strain OC-43 (HCoV OC-43) and human respiratory syncytial virus type 2 (HRSV-2) (enveloped RNA viruses), human herpes simplex virus type 1 (HSV-1) (enveloped DNA virus), human rhinovirus type 14 (HRV-14) (non-enveloped RNA virus) and human adenovirus type 5 (HadV-5) (non-enveloped DNA virus). The influence of the extracts on the internal replicative cycle of viruses was determined using the cytopathic effect (CPE) inhibition test. The virucidal activity, its impact on the stage of viral adsorption to the host cell, and its protective effect on healthy cells were evaluated using the final dilution method, making them the focal points of interest. The change in viral infectivity under the action of propolis extracts was compared with untreated controls, and Δlgs were determined. Most propolis samples administered during the viral replicative cycle demonstrated the strongest activity against HCoV OC-43 replication. The influence of propolis extracts on the viability of extracellular virions was expressed to a different degree in the various viruses studied, and the effect was significantly stronger in those with an envelope. Almost all extracts significantly inhibited the adsorption step of the herpes virus and, to a less extent, of the coronavirus to the host cell, and some of them applied before viral infection demonstrated a protective effect on healthy cells. Our results enlarge the knowledge about the action of propolis and could open new perspectives for its application in viral infection treatment.

## 1. Introduction

Propolis (bee glue) is a hydrophobic substance with sticky consistency produced by European honeybees (*Apis mellifera* L.), serving as a building and defensive material in their hives. Bees use propolis to smooth the internal walls of the hive, fill up cracks, repair and seal up the cells of the honeycomb, and embalm dead invaders inside the hive, thus removing the unpleasant smell and the microflora accompanying their decomposition and protecting the bee colony from infections [1,2]. In order to produce propolis, the worker bees collect resins from flowers and leaf buds of various plant species. Thereafter, they transport the material to the hive and mix it with beeswax and saliva secreted by their salivary glands. Some recent studies have revealed that the number of chemical compounds identified in propolis has reached 850. They form a heterogeneous mixture, which includes mainly polyphenolic compounds like flavonoids (quercetin, galangin, chrysin), aromatic acids and esters, aliphatic acids and esters, volatile compounds, waxy acids, carbohydrates, alcohols, aldehydes, ketones, steroids, enzymes, micro- and macronutrients, amino acids, vitamins, essential oils, pollen, and organic matter [3]. The physicochemical properties, phytochemical composition, and biological activities of propolis vary widely, which depends on the botanical source and the climatic characteristics of the geographic region from which it originates [4].

Propolis is one of the most valuable bee products, which has been used by humans for thousands of years as a remedy in folk and traditional medicine due to the lack of toxicity and the remarkable therapeutic properties. It has been found that the unique phytochemical composition of propolis determines its diverse pharmacological activities, such as antibacterial, antifungal, antiviral, antioxidant, immunomodulatory, antiparasitic, anti-allergic, anti-inflammatory, anticarcinogenic, anaesthetic, hepatoprotective, gastroprotective, anti-ulcerogenic, antidiabetic, astringent and other health beneficial effects [5,6].

Viral outbreaks are widely spread and represent an important problem for the health sector. The application of non-toxic natural products without adverse effects as alternatives of chemotherapeutics is essential for the therapy of viral infections. In this respect, propolis has shown great promise to be used as a potentially effective antiviral agent [7]. It has been reported to possess inhibitory activity against both DNA and RNA viruses. Amoros et al. (1992) [8] investigated the in vitro antiviral effect of propolis on herpes simplex virus type 1 and 2 (HSV-1 and HSV-2), adenovirus type 2, vesicular stomatitis virus and poliovirus type 2. The obtained results demonstrated that propolis had remarkable activity against poliovirus and herpes viruses, while vesicular stomatitis virus and adenovirus were less susceptible. Serkedjieva et al. (1992) [9] evaluated the in vitro antiviral activity of Bulgarian propolis on H3N2 and H1N1 influenza viruses and stated that propolis inhibited viral replication. Years later, the anti-influenza virus activity was confirmed by Kujumgiev et al. (1999) [10], who investigated the inhibitory effect of propolis extracts on avian influenza virus A/strain Weybridge (H7N7). Other studies reveal the antiviral potential of propolis against herpes simplex virus (HSV-1 and HSV-2) [11], coronavirus 2 (SARS-CoV-2) [12], varicella zoster virus [13], infectious bursal disease virus (IBDV), Reovirus [14] and canine distemper virus [15].

In the present study, the antiviral activity against structurally different viruses of six Bulgarian propolis extracts collected from six districts of Bulgaria was determined.

## 2. Materials and Methods

### 2.1. Host Cell Lines

The human colon carcinoma (HCT-8) cells were obtained from the American Type Culture Collection (ATCC) based in Manassas, VA, USA. The HCT-8 [HRT-18] cell line (ATCC-CCL-244, LGC Standards) was cultured in RPMI 1640 growth medium (ATCC-30-2001), supplemented with 10% horse serum (ATCC-30-2021), 0.3 g/L L-glutamine (Sigma-Aldrich, Darmstadt, Germany), and an antibiotic mixture of 100 IU penicillin and 0.1 mg streptomycin/mL (both from Sigma-Aldrich) at 37 °C with a constant supply of 5% CO_2_.

Madin–Darbey bovine kidney (MDBK) cells were provided by the National Bank for Industrial Microorganisms and Cell Cultures in Sofia, Bulgaria. These cells were grown in DMEM growth medium (Gibco, Grand Island, NY, USA) with 10% fetal bovine serum (Gibco BRL, USA), 10 mM HEPES buffer (AppliChem GmbH, Darmstadt, Germany), and antibiotics (100 IU/mL penicillin, 100 μg/mL streptomycin). The incubation occurred in a HERA cell 150 incubator (Heraeus, Hanau, Germany) at 37 °C with a 5% CO_2_ atmosphere.

The human epithelial type 2 (HEp-2) cells, derived from human laryngeal carcinoma, were obtained from the National Bank for Industrial Microorganisms and Cell Cultures in Sofia, Bulgaria. These cells were cultured in DMEM growth medium (Gibco, Grand Island, NY, USA) with 10% fetal bovine serum (Gibco, BRL, USA), 10 mM HEPES buffer (AppliChem GmbH, Darmstadt, Germany), and antibiotics (100 IU/mL penicillin, 100 μg/mL streptomycin). They were maintained in a HERA cell 150 incubator (Heraeus, Hanau, Germany) at 37 °C with a humidified atmosphere of 5% CO_2_.

Lastly, the human cervical epithelioid carcinoma cells (HeLa Ohio-I) were generously provided by Dr D. Barnard from Utah State University, Logan, UT, USA. These cells were cultured in DMEM growth medium (Gibco, Grand Island, NY, USA) with 10% fetal bovine serum (Gibco, BRL, USA), 10 mM HEPES buffer (AppliChem GmbH, Darmstadt, Germany), and antibiotics (100 IU/mL penicillin, 100 μg/mL streptomycin) at 37 °C with a 5% CO_2_ atmosphere in a HERA cell 150 incubator (Heraeus, Hanau, Germany).

### 2.2. Viruses

Human coronavirus OC-43 (HCoV-OC43) (ATCC: VR-1558) strain was cultured in HCT-8 cells using RPMI 1640 medium supplemented with 2% horse serum, 100 U/mL penicillin, and 100 μg/mL streptomycin. After 5 days of infection, cell lysis was performed through two freeze and thaw cycles, and the virus was titrated following the Reed and Muench formula. Both virus and mock aliquots were stored at −80 °C, as outlined in our previous study [16]. The infectious titer of the stock virus was found to be 10^6.5^ CCID_50_/mL.

Herpes simplex virus type 1, Victoria strain (HSV-1), obtained from Prof. S. Dundarov at the National Center of Infectious and Parasitic Diseases in Sofia, was replicated in confluent monolayers of MDBK cells using a maintenance solution Dulbecco’s modified Eagle medium (DMEM) from Gibco BRL, Paisley, Scotland, UK, supplemented with 0.5% fetal bovine serum (Gibco BRL, Scotland, UK), and antibiotics (100 IU/mL penicillin, 100 μg/mL streptomycin). Following incubation at 37 °C in a 5% CO_2_ incubator, the viral yield was frozen at −80 °C [16]. The infectious titer of the stock virus was determined to be 10^8.5^ CCID_50_/mL.

Human rhinovirus type 14 (strain 1059) (HRV-14) used for the experiments was purchased from the American Type Culture Collection (Manassas, VA, USA). HRV-14 stocks were prepared in HeLa Ohio-I cells using a maintenance DMEM medium with 2% fetal bovine serum and antibiotics (100 IU/mL penicillin, 100 μg/mL streptomycin). After incubation at 33 °C in a 5% CO_2_ incubator, the recovered virus was frozen at −80 °C. The stock virus titer was determined to be 10^3.5^ CCID_50_/mL.

Human respiratory syncytial virus type 2 (Long; HRSV-2), kindly provided by the Regional Center for Hygiene and Epidemiology, Plovdiv, Bulgaria, was grown in HEp-2 cells using DMEM maintenance medium (Gibco, BRL) containing 10 mmol/l HEPES buffer (Gibco, BRL), 0.5% fetal calf serum (Gibco BRL), and antibiotics (100 IU/mL penicillin, 100 μg/mL streptomycin). Following incubation at 37 °C in a 5% CO_2_ incubator, the viral yield was frozen at −80 °C. The infectious viral titer was determined to be 10^4.5^ CCID_50_/mL.

Human adenovirus type 5 (HadV-5), kindly provided by the District Center for Hygiene and Epidemiology, Plovdiv, Bulgaria, was replicated in HEp-2 cells in the presence of DMEM (Gibco, BRL) maintenance medium containing 10 mmol/L HEPES buffer (Gibco, BRL), 0.5% fetal calf serum (Gibco BRL), and antibiotics (100 IU/mL penicillin, 100 μg/mL streptomycin). The resulting amount of virus was frozen at −80 °C. The infectious viral titer was determined to be 10^5.0^ CCID_50_/mL.

### 2.3. Raw Propolis Material

Six fresh propolis samples collected from beekeepers at the end of the active beekeeping season (August–October) in six locations in Bulgaria were used in the study (Table 1). The samples were stored in plastic containers at room temperature in darkness until analysis.

### 2.4. Reference Compound

Remdesivir (GS-5734, RDV, REM, Veklury^®^) (Gilead Science Ireland UC) was initially dissolved in double distilled water to a concentration of 150 mg/mL and then diluted in RPMI nutrient medium to the required concentrations.

Acyclovir {ACV, [9-(2-hydroxyethoxymethyl)-guanine]} was kindly provided by the Deutsches Kresforschung Zentrum, Heidelberg, with a stock concentration of 3 mM solution in DMSO. Then, falling dilutions were made in DMEM medium to the required concentration.

Ribavirin (1-(β-D-ribofuranosyl)-1H-1,2,4-triazole-3-carboxamide), kindly provided by Prof. R. W. Sidwell, Utah State University, Logan, USA, was dissolved directly into the DMEM medium.

### 2.5. Preparation of Propolis Extracts

The raw propolis samples were finely ground using a blender (Bosch, Germany). The propolis extracts were prepared by weighing 1 g of sample and pouring 10 mL of 70% ethanol (Sigma-Aldrich, Merck, Germany) in a plastic tube. Next, the samples were left at room temperature for 72 h in darkness and periodically shaken on vortex V-1 (Biosan, Latvia) during the extraction period. The obtained extracts were filtered through filter paper and then stored at 4 °C for further analyses [17].

### 2.6. Total Phenolic Content

The total phenolic content (TPC) was determined by the standard method using a Folin–Ciocalteu reagent (Sigma-Aldrich, Merck), 1 mL of which was mixed with 0.8 mL of 7.5% sodium carbonate (Sigma-Aldrich, Merck) and 0.2 mL of the tested propolis extract. Then, the mixture was kept at room temperature for 20 min (in darkness), and the absorbance was measured at 765 nm (Camspec M107, Spectronic-Camspec Ltd., UK) against a blank (distilled water). The results were presented as mg equivalent of gallic acid (GAE)/g propolis [18].

### 2.7. Total Flavonoid Content

The total flavonoid content (TFC) was determined according to the standard procedure [18]. An aliquot of 1 mL of the tested propolis extract was mixed with 0.1 mL of 10% Al(NO_3_)_3_, 0.1 mL of 1 M CH_3_COOK, and 3.8 mL of distilled water. The sample was left at room temperature for 40 min, and then the absorbance was measured at 415 nm using quercetin as a standard. The results are expressed as mg quercetin equivalents (QE)/g propolis.

### 2.8. Antioxidant activity

DPPH radical scavenging assay. The reaction mixture containing 2.85 mL of DPPH reagent (2,2-diphenyl-1-picrylhydrazyl) and 0.15 mL of the tested propolis extract was incubated at 37 °C for 15 min. The reduction of absorbance was measured at 517 nm against a blank (methanol). The antioxidant activity was expressed as mM Trolox^®^ equivalents (TE)/g propolis [18].

Ferric-reducing antioxidant power (FRAP) assay. The FRAP reagent was freshly prepared with 300 mM acetate buffer with pH 3.6, 10 mM 2,4,6-Tris(2-pyridyl)-s-triazine (TPTZ) in 40 mM hydrochloric acid, and 20 mM Iron (III) chloride hexahydrate in distilled water in a ratio of 10:1:1. The reaction mixture (3 mL of FRAP reagent and 0.1 mL of the propolis extract) was incubated at 37 °C for 10 min, in darkness. The absorbance was measured at 593 nm against a blank (distilled water). The antioxidant activity was expressed as mM TE/g propolis [18].

### 2.9. Cytotoxicity Assay

A confluent monolayer of cell culture in 96-well plates (Costar^®^, Corning Inc., Kennebunk, ME, USA) was subjected to treatment with 0.1 mL/well of the support medium, either without the tested propolis extracts or with varying decreasing concentrations of the extracts. The cells were then incubated under specific conditions similar to those used for subsequent virus experiments: 33 °C with 5% CO_2_ for 5 days (for HCT-8), 33 °C with 5% CO_2_ for 2 days (for HeLa Ohio-I), and 37 °C with 5% CO_2_ for 2 days (for MDBK and HEp-2 cells). After the designated incubation period, the propolis extracts were removed, and the cells were washed before being incubated with neutral red (NR) dye at 37 °C for 3 h. The concentration of the test sample that reduced cell viability by 50% compared to untreated controls was defined as the 50% cytotoxic concentration (CC_50_). Each sample was tested in triplicate, with four wells per replicate.

The maximally tolerated concentration (MTC) of the extracts, which is the concentration at which they do not affect the cell monolayer, was also determined. The methodology is described in more detail in our previous study [16].

### 2.10. Antiviral Activity Assay

To assess the antiviral activity of propolis extracts, the cytopathic effect inhibition (CPE) test was employed. A 96-well plate with a confluent cell monolayer was infected with 100 cell culture infectious doses of 50% (CCID_50_) in 0.1 mL. After 2 h of adsorption at 33 °C (for HCoV OC-43 and HRV-14), 2 h of adsorption at 37 °C (for HRSV-2) and 1 h of adsorption at 37 °C (for HSV-1 and HadV-5) unattached virus was removed, and the tested extract was added at different concentrations, and the cells were incubated for 5 days at 33 °C (for HCoV OC-43); 2 days at 33 °C (for HRV-14) or 2 days at 37 °C (for HRSV-2, HSV-1 and HadV-5) and in the presence of 5% CO_2_. The cytopathic effect was determined using a neutral red uptake assay, and the percentage of CPE inhibition for each test sample concentration was calculated using the following formula:% CPE = [OD test sample − OD virus control]/[OD toxicity control − OD virus control] × 100
where ODtest sample is the mean of the ODs of the wells inoculated with the virus and treated with the test sample at the corresponding concentration, ODs virus control is the mean of the ODs of the virus control wells (no compound in the medium). OD control for toxicity is the mean of the ODs of the wells not inoculated with the virus but treated with the corresponding concentration of the test compound. The 50% inhibitory concentration (IC_50_) is defined as the concentration of the test substance that inhibits 50% of viral replication compared to the viral control. The selectivity index (SI) is calculated from the CC_50_/IC_50_ ratio [16].

### 2.11. Virucidal Assay

Preparations were made with a total volume of 1 mL containing virus (104 CCID_50_) and the tested propolis extract at its maximum permissible concentration (MTC) in a 1:1 ratio. In parallel, a sample was created with untreated virus diluted 1:1 with DMEM medium. Both the control and experimental samples were incubated at room temperature for various time intervals (15, 30, 60, 90, and 120 min). Using the endpoint dilution method of Reed and Muench (1938) [19], the residual infectious virus content in each sample and the Δlgs compared to untreated controls were then determined.

### 2.12. Effect on Viral Adsorption

To initiate the experiments, twenty-four well plates were chilled to 4 °C and then inoculated with 10^4^ CCID^50^ of either HCoV OC-43 or HSV-1, depending on whether HCT-8 or MDBK monolayers were used, respectively. Concurrently, the monolayer was treated with the tested propolis extracts at their maximum permissible concentration (MTC) and kept at 4 °C during the virus adsorption time. The virus and propolis extract were removed at different time intervals, which varied for the two types of viruses (15, 30, 45, and 60 min for HSV-1, and 15, 30, 60, 90, and 120 min for HCoV OC-43). Subsequently, the cells were washed with PBS and covered with maintenance medium before being incubated at 37 °C (for HSV-1) or 33 °C (for HCoV OC-43) in the presence of 5% CO_2_ for 24 h. After three cycles of freezing and thawing, the infectious viral titer of each sample was determined and compared to the viral titer of the control for the respective time interval. The Δlgs (logarithmic differences) were then calculated. Each sample was prepared in quadruplicate for reliable data analysis.

### 2.13. Pre-Treatment of Healthy Cells

Previously grown monolayers of MDBK or HCT-8 cells in 24-well cell culture plates (CELLSTAR, Greiner Bio-One) were exposed to the propolis extracts at their maximum permissible concentration (MTC). The samples were then incubated at 37 °C for different time intervals of 15, 30, 60, 90, and 120 min. After the designated time, the extracts were removed, and the cells were washed with PBS before being inoculated with the respective virus strain (1000 CCID_50_ in 1 mL/well). For HCoV OC-43, the virus adsorption lasted for 120 min, and for HSV-1, it was 60 min. Afterwards, any unadsorbed virus was removed, and the cells were covered with a support medium. Subsequently, the samples were incubated at 33 °C (for HCoV OC-43) or 37 °C (for HSV-1) in the presence of 5% CO_2_ for 24 h. Following this incubation period, the samples were subjected to triplicate freezing and thawing, and the infectious virus titers were determined. Δlg (logarithmic differences) were calculated by comparing the viral titer of the treated samples to the viral titer of the control (untreated with extract) for the respective time interval. Each sample was prepared in quadruplicate for accurate and reliable analysis.

### 2.14. Statistical Analysis

Data on cytotoxicity and antiviral effects were analysed statistically. The values of CC_50_ and IC_50_ were presented as means ± SD. The differences’ significance between the cytotoxicity values of propolis extracts and the reference substances, as well as between the effects of the test products on the viral replication, was performed by Student’s *t*-test, with *p*-values of <0.05 were considered significant. The final data sets were analysed with the Graph Pad Prism 4 software.

## 3. Results

From the studies carried out so far on the composition of various types of propolis, it has been established that its composition includes about 850 ingredients. The main compounds that contribute to its biological activities are polyphenols and especially flavonoids [3]. Therefore, we focused our attention on the ingredients contained in the propolis extracts we studied. The propolis extracts presented in this manuscript were selected after an initial screening selection. Eighty propolis extracts from different territories of Bulgaria were studied. Of all the extracts, these six showed the highest content of polyphenols and flavonoids, as well as the most distinct antioxidant activity. Table 2 presents the results of total phenolic content (TPC) and total flavonoid content (TFC) of the six propolis extracts. Of the studied samples, PS6 showed the highest amount of TPC (256.1 ± 0.56 mg GAE/g propolis), and PS5 demonstrated the lowest (151.7 ± 0.32 mg GAE/g propolis). Comparing the obtained data for TFC, the highest value was reported for PS2 (124.1 ± 2.23 mg QE/g propolis) and the lowest for PS6 (74.6 ± 0.15 mg QE/g propolis).

Antioxidant activity results were evaluated by two methods. The DPPH assay showed the highest antioxidant activity in PS2, while PS1 had the lowest antioxidant potential of the studied propolis extracts. According to the FRAP method, PS6 possessed the highest antioxidant activity, and PS5 showed the lowest value by the same method (Table 2).

In order to avoid the negative influence of toxic concentrations of the propolis extracts when conducting the antiviral experiments, the cytotoxicity they exert on the cells was determined in advance. The effect of the extracts on the cells was determined against the four cell lines on which the antiviral experiments were carried out in the next step (HCT-8, MDBK, HEp-2 and HeLa Ohio cells). Against MDBK, HEp-2 and HeLa Ohio cell lines, cytotoxicity was measured after two days of incubation with the extracts, as this is the time interval at which antiviral experiments are determined for the respective viruses. To achieve a good cytopathic effect with HCoV OC-43, a longer time of 5 days was required. Therefore, cytotoxicity with the HCT-8 cell line was also measured on day 5.

From the experiments performed, it can be seen that, in general, the highest cytotoxicity was reported for the HCT-8 cell line, most likely because the exposure time was the longest. From the rest of the cell lines, where the effect was measured for the same time interval, the extracts showed the weakest cytotoxicity against the HeLa Ohio cell line, close but slightly higher toxicity in MDBK cells, and the most sensitive to the action of propolis extracts from the cells tested turned out to be HEp-2. The weakest cytotoxicity was shown by the extracts PS4, PS3 and PS1 against HeLa Ohio cells, as well as PS1 and PS3 against the MDBK cell line. As a general effect on the four cell lines, PS6 showed the highest toxicity. When comparing the cytotoxicity of the tested samples to the reference substances used, it is noticed that they are less toxic than Ribavirin but demonstrate several times higher cytotoxicity compared to Acyclovir and Remdesivir (Table 3).

Once the non-toxic concentration range of the propolis samples was identified, their impact on the internal replicative cycle of the virus was studied at concentrations below the CC_50_. In general, the influence of the extracts is strongest for the herpes virus and to a slightly lesser extent for the coronavirus and the rhinovirus. The effect was weakest with adenovirus, and almost in the same range was the inhibition reported with RSV-2. The highest selective index (SI) of all propolis samples showed PS4 against HSV-1 (SI = 45.3) and HCoV OC-43 (SI = 43.3). PS2 also showed significant activity against HSV-1 replication (SI = 32.9). The distinct activity was also demonstrated by PS1 (SI = 28.7) applied to the replication of HRV-14 and PS2 (SI = 26.6), affecting HCoV OC-43. The substances PS4 (SI = 22.3) for HRV-14 and PS6 (SI = 21.7) for HSV-1 have similar activity. Significantly lower was the influence of PS2 (SI = 10.0) on HRV-14; PS5 (SI = 8.9) in HSV-1; and PS 1 and PS2 with SI between 7.5 and 8.6 for RSV-2 and HAdV-5 (Table 4).

Having assessed the effect of the investigated propolis samples on the replication of structurally diverse viruses, the subsequent phase of our research involved examining the impact of the extracts on the vitality of extracellular virions. The results from the experiments showed a stronger effect on enveloped viruses compared to non-enveloped ones. The effect was monitored at different time intervals, and, in general, a dependence of the effect on the exposure time was noticed. With a longer exposure, the inhibition of virus particles increased. The most significant was the effect of PS5 (Δlg = 2.25) at 90 and 120 min on HCoV OC-43 virions, with a similar effect on HSV-1 and, to a less extent, on HRSV-2. A distinct effect of Δlg = 2.0 in HCoV OC-43 was also demonstrated by PS4 and PS6, whose influence on HRSV-2 was less pronounced, respectively, Δlg = 1.6 and Δlg = 2.0 at the longest interval of 120 min. PS6 was the only one of the six investigated extracts which exhibited a significant effect (Δlg = 1 = 8) at 120 min on HRV-14. PS1 also exerted a suppressive effect on extracellular HSV-1, even at a contact duration interval of 30 min; compared to HCoV OC-43 virions, the effect was weaker (Δlg = 1.75) (Table 5 and Table 6).

Having determined the effect of propolis extracts on extracellular virions and virus replication in the cell, the next step in our research was to follow the effect of the extracts on the adsorption step of the virus to the cell. The experiments were carried out with the two viruses for which we obtained the most distinct activity so far—HSV-1 and HCoV OC-43. The influence was again followed at different time intervals depending on the duration of viral adsorption (up to 60 min for the herpes virus and up to 120 min for the coronavirus). All extracts demonstrated varying degrees of influence on this stage of viral reproduction. The inhibition of the process was more pronounced in HSV-1 compared to HCoV OC-43. The strongest effect on HSV-1 was shown by PS5 and PS6 (Δlg = 2.25) 30 min after exposure, and the influence of PS6 was significant as early as 15 min (Δlg = 1.75). The effect of PS2, PS3 and PS4 (Δlg = 2.0) was also significant at 30 min and remained unchanged until the last investigated time interval of 60 min. PS1, although to a less extent, also affected the adsorption stage of the virus with a decrease in the viral titer with Δlg = 1.75.

When tracking the adsorption of HCoV OC-43 to the HCT-8 cells, a significantly weaker influence of the extracts was observed, and at a slightly longer contact period—at 90 or 120 min. Here, the influence of PS4 is most significant per 120 min (Δlg = 2.25). Additionally, a distinct effect at 120 min of Δlg = 1.75 was demonstrated by PS1, PS2 and PS5. PS3 and PS6 showed weak activity towards the adsorption of HCoV OC-43 (Table 7).

After assessing the influence of propolis extracts on various stages of viral reproduction and vitality, we investigated whether these substances provided a protective effect on healthy cells, guarding them against subsequent viral infections. The experiments were once again conducted using the MDBK and HCT-8 cell lines, along with the viral strains HSV-1 and HCoV OC-43. The results revealed that none of the extracts significantly protected HCT-8 cells from HCoV OC-43 infection, with the maximum decrease in viral titer being Δlg = 1.0. However, in the case of MDBK cells, the effect was notably stronger. The most pronounced protection was observed with PS3 after 15 min of treatment (Δlg = 2.0), and this effect further increased to Δlg = 2.25 after 120 min. Similar protection was shown by PS4, but the effect was significant at 30 min. PS2 also has a strong influence, which maintains the same activity during all monitored time intervals (Δlg = 2.0). The other three propolis extracts: PS1, PS5 and PS6, showed weak protective effects on sensitive healthy cells (Table 8).

## 4. Discussion

In some viruses that cause respiratory, intestinal, skin, and sexually transmitted infections in humans, antiviral agents have been developed that significantly reduce the severity of symptoms and shorten the recovery period. However, therapy failure is increasingly observed due to the selection of therapy-resistant mutants [21]. This is a major reason for the intensive search for new, unconventional antiviral agents closer to the cell components, are low toxic, cause fewer side effects, and can serve as an alternative to the currently used antiviral therapeutics.

In recent decades, more and more data have been accumulated from the study of the various biological activities of propolis. One of its benefits is the impact on developing viral infections [22,23]. The exact mechanisms by which its influence is carried out are still not sufficiently studied. Two main directions of its action have been established: (1) direct interaction with the virus or on stages of its replication [13,23,24,25,26,27] and (2) stimulation of the immune system to overcome infection [28,29,30].

The mechanism and strength of the antiviral effect of propolis are determined by the multitude of substances included in its composition, which is determined by the geographical location and the season of the year in which it was obtained [30]. In the temperate zones of Europe, North America and Asia, where the predominant source is poplar tree species, most often black poplar (*Populus nigra*), propolis contains mainly flavanones and flavones and smaller amounts of phenolic acids and their esters [31]. Propolis flavonoids show antiviral activity against DNA and RNA viruses [32] and immunomodulatory action [30]. Over 150 types of flavonoids have been found in different types of propolis [33]. Propolis from tropical countries contains mainly complex phenolic compounds such as prenylated para-coumaric acids, prenylated flavonoids, caffeoylquinic acid derivatives and lignans [34].

Some studies show that propolis can affect virus replication [28] by reducing the synthesis of viral RNA transcripts in cells and thus reducing the number of coronavirus particles [13] or by inhibiting Varicella zoster virus DNA polymerase [27]. Another potential mechanism of inhibition of viral replication is the proven inhibitory activity of Sulawesi propolis compounds against the enzymatic activity of SARS-CoV-2 main protease [25].

Much research has shown that in contact with the viral particle, propolis destroys the ability of the pathogen to enter the cell [28,35,36]. Virus particles with altered morphology were observed, suggesting possible damage to viral envelope proteins. Virions were also found in an electrodense layer formed around the cell membrane. This has been suggested to affect the entry of the virus into the host cell and disrupt its replication cycle [37].

Basically, viruses are divided into two groups—enveloped and non-enveloped. Non-enveloped viruses are covered with specific viral proteins (capsomeres) (forming the viral capsid), which can be the target of the action of various substances outside the cell. In most cases, these proteins are stable and difficult to influence. Among the viruses we use with such a structure are human rhinovirus type 14 (HRV-14) and human adenovirus type 5 (HadV-5). In enveloped viruses, there is another shell on top of the capsid, which comprises lipids and proteins. This envelope is much more sensitive to the action of different substances, so it has been experimentally shown that when the same substance acts on an enveloped and a non-enveloped virus, the effect on the enveloped virus is significantly stronger. Among the enveloped viruses used in our study are: human coronavirus strain OC-43 (HCoV OC-43), human respiratory syncytial virus type 2 (HRSV-2), and human herpes simplex virus type 1 (HSV-1). Our studies on the virucidal activity of propolis extracts prove the observation described above.

After entering the host cell, each type of virus has its specific viral enzymes, thanks to which it manages to build many of its structural components and assemble them into new virus particles that leave the host cell. These specific viral enzymes differ in a number of characteristics from cellular ones and are putative targets for attack by structural components contained in propolis extracts. As a result of such an interaction, the processes of transcription, translation and replication of viral components and/or their assembly and exit from the host cell are disrupted.

Our results were in agreement with other scientific publications on this topic. We used different experimental setups, each adding the propolis extracts at different stages of the viral infection cycle. In a similar way, the data obtained by other researchers were close to ours, proving a significant influence of propolis on extracellular virions, especially in enveloped viruses, as well as on the stage of viral adsorption on susceptible cells [35,36,38]. A more detailed study of the specific mechanism of action will take place after a more detailed study of the chemical composition of the samples.

We obtained similar results in our previous study of Canadian propolis [39], where we demonstrated an effect on virions and the adsorption stage of Herpes simplex virus types 1 and 2 to MDBK cells. In the present study with the Bulgarian propolis, an influence on the viral replicative cycle was also found in some of the samples, which is clearly due to the differences in the composition of the propolis due to their different geographical origin.

The main belief among human society is that propolis acts on the immune system and thus helps the body overcome various infectious diseases. Our research proves that propolis also has an inhibitory effect on various viruses, directly damaging their structure or affecting stages of their replication.

We also introduced a new research setup where we treated the still healthy cells with the propolis extracts and determined the degree of protection that propolis shows on the cell membrane from subsequent viral infection. Our results reconfirm the data obtained by other teams who used a similar experimental methodology and found that the application of 0.5 mg/mL EEP two hours before infection in MDBK cells caused a reduction in the number of Aujeszky’s disease virus formed plaque compared to the other treatments used or to the infected and untreated culture [37].

This proves that if propolis is used prophylactically, it could significantly protect healthy cells in our body from viral invasion. In combination with its already proven antiviral and immunomodulating activities, its application, especially in periods of epidemics, would reduce morbidity and shorten the recovery period.

## 5. Conclusions

The present study once again confirms our previously reported data on the antiviral activity of propolis. The study of the activity against the replication of structurally distinct viruses showed a different degree of inhibition in individual propolis samples. In direct contact of the propolis extracts with the virus particles, it was found that the effect was stronger for enveloped viruses, which is most likely the result of the interaction of the components included in the composition of propolis with the viral proteins of the envelope, necessary for attachment and entry into the cell, resulting in inactivation of the virus. The data accumulated so far on the activity of propolis present it as a promising candidate for inclusion in the prevention and treatment of many infectious diseases. Still, it is necessary to expand the knowledge of its mechanism of action for its more complete application.

## Figures and Tables

**Table 1 life-13-01611-t001:** Origin of the propolis samples.

Propolis Sample (PS)	Town/Village	Municipality	District	GPS Coordinates
PS1	Silistra	Silistra	Silistra	44°07′ N 27°17′ E
PS2	Simitli	Simitli	Blagoevgrad	41°54′ N 23°08′ E
PS3	Gorna Malina	Gorna Malina	Sofia	42°41′ N 23°42′ E
PS4	Shumen	Shumen	Shumen	43°16′ N 26°55′ E
PS5	Vladimir	Radomir	Pernik	42°26′ N 23°05′ E
PS6	Cherven breg	Dupnitsa	Kyustendil	42°18′ N 23°10′ E

**Table 2 life-13-01611-t002:** Total phenolic content (TPC), total flavonoid content (TFC) and antioxidant activity of the ethanolic propolis extracts.

Propolis Sample	TPC, mg GAE/g	TFC, mg QE/g	Antioxidant Activity	
			DPPH, mM TE/g	FRAP, mM TE/g
PS1	216.9 ± 0.28	78.3 ± 0.38	873.2 ± 14.50	867.5 ± 60.00
PS2 *	168.0 ± 0.63	124.1 ± 2.23	1201.4 ± 26.23	715.3 ± 6.12
PS3	243.1 ± 0.28	87.9 ± 0.11	1106.9 ± 43.50	1000.1 ± 37.50
PS4	230.8 ± 0.30	77.0 ± 0.50	1083.3 ± 34.80	978.9 ± 17.50
PS5 *	151.7 ± 0.32	87.0 ± 2.01	1016.3 ± 22.20	645.3 ± 11.25
PS6	256.1 ± 0.56	74.6 ± 0.15	1133.5 ± 23.22	1085.0 ± 22.50

* The result is presented in our previous study [20].

**Table 3 life-13-01611-t003:** In vitro assessment of cytotoxicity of the propolis extracts.

Propolis Sample	Cytotoxicity (μg/mL)							
	HCT-8		MDBK		HEp-2		HeLa Ohio	
	CC_50_	MTC	CC_50_	MTC	CC_50_	MTC	CC_50_	MTC
PS1	58.0 ± 8.2 **	10.0	120.0 ± 6.7 **	32.0	56.5 ± 2.1 *	30.0	158.0 ± 7.2 **	70.0
PS2	48.0 ± 4.5 **	10.0	72.5 ± 3.8 **	10.0	68.2 ± 3.3 *	30.0	77.7 ± 2.9 *	35.0
PS3	62.6 ± 5.2 **	10.0	104.7 ± 7.3 **	32.0	57.0 ± 2.5 *	30.0	174.0 ± 6.5 **	70.0
PS4	52.0 ± 3.6 **	10.0	73.5 ± 3.2 **	10.0	67.7 ± 2.9 *	30.0	190.0 ± 7.0 **	70.0
PS5	59.8 ± 7.3 **	10.0	71.8 ± 4.7 **	10.0	62.5 ± 2.4 *	30.0	77.0 ± 2.5 *	35.0
PS6	57.0 ± 6.5 **	10.0	66.2 ± 5.7 **	10.0	59.4 ± 1.5 *	20.0	58.0 ± 2.0 *	30.0
Acyclovir	nd	nd	291.0 ± 9.4 **	nd	nd	nd	nd	nd
Remdesivir	250.0 ± 4.3	nd	nd	nd	nd	nd	nd	nd
Ribavirin	nd	nd	nd	nd	14.0 ± 0.5	nd	34.0 ± 0.5	nd

nd—no data; * *p* < 0.05; when comparing the value of each propolis extract with the corresponding reference substance for the given cell line; ** *p* < 0.001 when comparing the value of each propolis extract with the corresponding reference substance for the given cell line.

**Table 4 life-13-01611-t004:** In vitro antiviral activity of the propolis extracts.

Propolis Sample	Antivirus Activity (μg/mL)									
	HCoV OC-43		HSV-1		HAdV-5		RSV-2		HRV-14	
	IC_50_ (μg/mL)	SI	IC_50_ (μg/mL)	SI	IC_50_ (μg/mL)	SI	IC_50_ (μg/mL)	SI	IC_50_ (μg/mL)	SI
PS1	-	-	-	-	7.5 ± 0.2 **	7.5	6.7 ± 0.3 **	8.4	5.5 ± 0.2 **	28.7
PS2	1.8 ± 0.3 *	26.6	2.2 ± 0.3 *	32.9	8.2 ± 0.3 **	8.3	7.9 ± 0.3 **	8.6	7.8 ± 0.2 **	10.0
PS3	-	-	-	-	-	-	47.0 ± 2.2 **	1.2	7.0 ± 0.1 **	2.8
PS4	1.2 ± 0.1 *	43.3	1.4 ± 0.02 *	45.3	-	-	44.0 ± 2.4 **	1.5	8.5 ± 0.3 **	22.3
PS5	10.2 ± 0.9	5.8	8.2 ± 1.2 **	8.9	-	-	46.0 ± 1.8 **	1.4	-	-
PS6	10.8 ± 0.7	5.2	3.3 ± 0.8 **	21.7	-	-	-	-	-	-
Acyclovir	nd	nd	0.33 ± 0.03	881.8	nd	nd	nd	nd	nd	nd
Remdesivir	12.5 ± 0.9	200.0	nd	nd	nd	nd	nd	nd	nd	nd
Ribavirin	nd	nd	nd	nd	0.2 ± 0.01	70.0	0.3 ± 0.01	46.6	0.5 ± 0.02	68.0

-, lack of inhibition of viral replication; nd—no data; *****
*p* < 0.05; when comparing the value of each propolis extract with the corresponding reference substance for the given virus strain; ******
*p* < 0.001, when comparing the value of each propolis extract with the corresponding reference substance for the given virus strain.

**Table 5 life-13-01611-t005:** Virucidal activity of enveloped viruses.

Propolis Sample	Δlg														
	HCoV OC-43					HRSV-2					HSV-1				
	15 min	30 min	60 min	90 min	120 min	15 min	30 min	60 min	90 min	120 min	15 min	30 min	60 min	90 min	120 min
PS1	1.25	1.25	1.25	1.75	1.75	0.0	0.0	0.5	1.0	1.0	1.0	1.75	1.75	2.0	2.0
PS2	1.0	1.25	1.5	1.5	1.5	0.2	0.2	0.6	1.5	1.5	1.0	1.0	1.5	1.5	1.5
PS3	0.5	0.5	1.0	1.5	1.5	0.2	0.2	0.7	1.2	1.6	1.0	1.0	1.25	1.25	1.25
PS4	0.5	1.0	1.5	2.0	2.0	0.3	0.3	0.7	1.4	1.6	1.0	1.0	1.5	1.5	1.5
PS5	1.25	1.25	1.25	2.25	2.25	0.3	0.3	0.9	1.5	2.0	1.0	1.0	1.5	1.75	2.25
PS6	1.0	1.25	1.25	2.0	2.0	0.2	0.2	0.9	1.7	2.0	1.0	1.0	1.5	1.5	1.75
70% etanol	5.75	5.75	5.75	5.75	5.75	4.5	4.5	4.5	4.5	4.5	4.75	4.75	4.74	4.74	4.74

**Table 6 life-13-01611-t006:** Virucidal activity of non-enveloped viruses.

Propolis Sample	Δlg									
	HAdV-5					HRV-14				
	15 min	30 min	60 min	90 min	120 min	15 min	30 min	60 min	90 min	120 min
PS1	0.0	0.0	0.5	1.0	1.0	0.0	0.0	0.1	1.4	1.6
PS2	0.0	0.0	0.7	1.0	1.0	0.0	0.0	0.0	1.2	1.5
PS3	0.0	0.0	0.5	1.0	1.0	0.0	0.0	0.3	1.2	1.6
PS4	0.0	0.0	0.8	0.8	1.2	0.0	0.0	0.2	1.3	1.5
PS5	0.0	0.0	0.4	0.8	0.8	0.0	0.0	0.2	1.3	1.3
PS6	0.0	0.0	0.6	0.8	1.0	0.0	0.0	0.0	1.5	1.8
70% etanol	5.0	5.0	5.0	5.0	5.0	3.2	3.2	3.2	3.2	3.2

**Table 7 life-13-01611-t007:** Influence of the extracts on the stage of adsorption of HSV-1 and HCoV OC-43 to sensitive cells.

Propolis Sample	Δlg								
	HSV-1				HCoV OC-43				
	15 min	30 min	45 min	60 min	15 min	30 min	60 min	90 min	120 min
PS1	1.5	1.75	1.75	1.75	1.0	1.0	1.0	1.5	1.75
PS2	1.5	2.0	2.0	2.0	1.0	1.0	1.0	1.25	1.75
PS3	1.5	1.75	2.0	2.0	0.75	1.0	1.0	1.25	1.5
PS4	1.5	2.0	2.0	2.0	0.75	0.75	1.0	1.75	2.25
PS5	1.25	2.25	2.25	2.25	1.5	1.5	1.5	1.75	1.75
PS6	1.75	2.25	2.25	2.25	1.5	1.5	1.5	1.5	1.5

**Table 8 life-13-01611-t008:** Protective effect of pre-treatment of extracts on healthy cells and subsequent virus infection.

Propolis Sample	Δlg									
	HSV-1					HCoV OC-43				
	15 min	30 min	60 min	90 min	120 min	15 min	30 min	60 min	90 min	120 min
PS1	1.25	1.25	1.25	1.25	1.25	0.5	0.5	0.5	1.0	1.0
PS2	2.0	2.0	2.0	2.0	2.0	0.5	0.5	0.5	1.0	1.0
PS3	2.0	2.0	2.0	2.0	2.25	0.5	0.5	0.5	1.0	1.0
PS4	1.0	2.0	2.0	2.0	2.25	0.5	0.5	0.5	1.0	1.0
PS5	1.0	1.0	1.0	1.0	1.0	1.0	1.0	1.0	1.0	1.0
PS6	1.0	1.0	1.0	1.0	1.25	0.5	0.5	0.5	0.5	0.5

## Data Availability

Not applicable.

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
