# Peer review of "Antiviral Potential of Specially Selected Bulgarian Propolis Extracts: In Vitro Activity against Structurally Different Viruses"

_life, 2023, doi:10.3390/life13071611_

Round 1
Reviewer 1 Report
- Rewrite the discussion section.
- Line 21: "Propolis didn't have hormones, revised.
- Some incorrect references need to be updated.
- Abbreviations should be clear when they appear for the first time.
- A few typos need to be fixed.
Author Response
Response to Reviewer 1 Comments
- The discussion section has been revised.
- The expression was corrected.
- We checked the references, but did not find any that the respected reviewer considered incorrect.
- Thank you for the remark. I actually found an untyped abbreviation. It is supplemented in the manuscript.
Reviewer 2 Report
Dear Editor,
I have read the manuscript titled "Antiviral potential of specially selected Bulgarian propolis extracts: in vitro mechanism of action against structurally different viruses.” As stated by the authors, the study aimed to investigate the antiviral activity against structurally different viruses of six Bulgarian propolis extracts collected from six districts of Bulgaria. The authors concluded that their research enlarged the knowledge about the action of propolis and could open new perspectives for its application in viral infection treatment.
The routine employed procedures were well-executed. The manuscript is well-organized, and the results are interesting, but they need more integration in the discussion. Unfortunately, I could not detect an outstanding contribution that advances in the field of the biological activity of propolis. As mentioned before, I consider the manuscript unsuitable for publication in its present form.
To consider the publication of the manuscript, the author needs to address the following issues:
1. The authors must include a more detailed chemical composition and spectral characteristics of the propolis used. In the manuscript, it has been assumed that the biological activity displayed is due to its phenol and flavonoid content. However, it should be noted that numerous other components in propolis may act as synergists or even as antagonists.
2. Regarding the previous point, the authors should mention in the introduction the specific properties or features of the chosen Bulgarian propolis that lead them to believe it can effectively combat the targeted viruses.
3. The authors must provide a detailed description of the structural features of the viruses that render them vulnerable to the chosen propolis.
4. The authors need to provide further explanation regarding their analysis of the antiviral action mechanism of the chosen propolis. Their experiments only covered various stages of the viral infection but have yet to learn about the mechanism displayed in each stage.
5. The discussion section must be improved to highlight this paper’s results or contribution to the field.
6. I noticed numerous “no data” instances in the result section. Could the authors please clarify the reason behind this? It would also be helpful to know the highest concentration tested for context
The English language used in the manuscript is of good quality, but there are a few typographical errors and instances of word repetition that could be improved.
Author Response
Response to Reviewer 2 Comments
1.As we indicated in the manuscript, propolis contains about 850 chemical compounds. The majority of them are polyphenols and flavonoids, so in this study we have stopped our attention only on the content of these groups of compounds. On the other hand, it is known that the composition of propolis is not constant. It varies depending on the geographical location from where it is extracted. It was also established that in different years in the same locality propolis has a different composition, as well as that there are differences in composition in the different months of the year during which it can be collected. Therefore, the determination of a more precise chemical composition of the studied propolises is not the aim of the present study. A similar study is planned for our future research, where we will probably also compare the chemical composition of propolises collected from the same beehives, but from different years.
- The propolis extracts presented in this manuscript were selected after an initial screening selection. 80 propolis extracts from different territories of Bulgaria were studied. Of all the extracts, these six showed the highest content of polyphenols and flavonoids, as well as the most distinct antioxidant and antibacterial activity. We have not presented these studies of ours because they are included in other manuscripts that have not yet been accepted for publication and we cannot cite them. But these studies were the reason for selecting exactly these 6 extracts from the original 80, to study also their antiviral activity, which, as seen from our study, is different for structurally different viruses.
- Basically, viruses are divided into two groups - enveloped and non-enveloped. Non-enveloped viruses are covered on the outside with specific viral proteins (capsomeres) (forming the viral capsid), which can be the target of the action of various substances outside the cell. In most cases, these proteins are stable and difficult to influence. Among the viruses we use with such a structure are: Human rhinovirus type 14 (HRV-14) and Human adenovirus type 5 (HadV-5). In enveloped viruses, there is another shell on top of the capsid, which is made up of lipids and proteins. This envelope is much more sensitive to the action of different substances, so it has been experimentally shown that when the same substance acts on an enveloped and an non-enveloped virus, the effect on the enveloped virus is significantly stronger. Among the enveloped viruses used in our study are: Human coronavirus strain OC-43 (HCoV OC-43), Human respiratory syncytial virus type 2 (HRSV-2) and Human herpes simplex virus type 1 (HSV-1). Our studies on the virucidal activity of propolis extracts prove the observation described above.
After entering the host cell, each type of virus has its specific viral enzymes, thanks to which it manages to build many of its structural components, assemble them into new virus particles that leave the host cell. These specific viral enzymes differ in a number of characteristics from cellular ones and are putative targets for attack by structural components contained in propolis extracts. As a result of such an interaction, the processes of transcription, translation and replication of viral components and/or their assembly and exit from the host cell are disrupted.
- Indeed, the present study only investigated the effect of propolis extracts on different stages of viral infection. A more detailed study of the specific mechanism of action will take place after a more detailed study of the chemical composition of the samples. As the esteemed reviewer mentioned earlier, the action of propolis extracts is the result of the synergistic, and possibly antagonistic action of the components contained in it.
- The discussion section has been revised.
- In the "Results" section, we have used the expression “no data”, which is used only for the reference substances acyclovir, remdesivir and ribavirin. Acyclovir is a reference substance in herpes infection. Therefore, it has not been studied in the other types of infections and we have indicated that "no data". Similarly, remdesivir is only for the coronavirus infection, and ribavirin for the rhino-, adeno- and human respiratory virus. Since each virus replicates only in a specific cell type, the corresponding reference substance was only tested on the specific cell line, and again "no data" for the others.
Reviewer 3 Report
Please correct the remarks in manuscript

Moderate editing of the English langue
Author Response
Response to Reviewer 3 Comments
We thank the esteemed reviewer for the remarks made. Corresponding corrections have been added to the manuscript.
Round 2
Reviewer 1 Report
Yes authors modified the manuscript according to our comments
Author Response
Thanks for the support and recommendations!
Reviewer 2 Report
Dear Editor,
I have reviewed the improved manuscript "Antiviral Potential of Bulgarian Propolis Extracts: In Vitro Mechanism of Action Against Different Viruses". While I noticed that some comments had been addressed, many were still missing from the manuscript. I recommend that the authors include the answers to comments 2, 3, and 4 in the manuscript to enhance its different sections. Furthermore, although the author mentioned that the discussion section had been improved, I did not see any highlighted text indicating this work's contribution.
The authors responded to comment 4 by stating that the study only investigated the effect of propolis extracts on different stages of viral infection. A more detailed study of the specific mechanism of action will take place after a more detailed study of the chemical composition of the samples. The manuscript title must be modified to reflect this.
After the modifications described above, the manuscript can be accepted for publication.
The quality of the English language is adequate and only requires moderate editing.
Author Response
I thank the respected reviewer for his comments and suggestions, I believe they will improve the quality of our manuscript.
- At the suggestion of the reviewer, the answers from the first round of questions 2, 3 and 4 have been entered into the manuscript.
- I apologize for the missing corrections in the discussion after the first round of reviews. Indeed, I had made corrections that I now do not see in the manuscript. I entered them again.
- We made a small change to the title. I hope this is more correct.
Reviewer 3 Report
Ok corrected

Author Response

(The authors gave the same response as above.)
